# A Time-Based Objective Measure of Exposure to the Food Environment

**DOI:** 10.3390/ijerph16071180

**Published:** 2019-04-02

**Authors:** Jason Y. Scully, Anne Vernez Moudon, Philip M. Hurvitz, Anju Aggarwal, Adam Drewnowski

**Affiliations:** 1Department of Planning and Public Administration, Eastern Washington University, 668 N Riverpoint Blvd, Suite A, Spokane, WA 99202, USA; jscully@ewu.edu; 2The Urban Form Lab, University of Washington, Seattle, WA 98195, USA; moudon@uw.edu; 3Center for Public Health Nutrition, University of Washington, Seattle, WA 98105, USA; anjuagg@uw.edu (A.A.); adamdrew@uw.edu (A.D.)

**Keywords:** Fast food, spatio-temporal exposure, mobility patterns, GPS, selective mobility bias

## Abstract

Exposure to food environments has mainly been limited to counting food outlets near participants’ homes. This study considers food environment exposures in time and space using global positioning systems (GPS) records and fast food restaurants (FFRs) as the environment of interest. Data came from 412 participants (median participant age of 45) in the Seattle Obesity Study II who completed a survey, wore GPS receivers, and filled out travel logs for seven days. FFR locations were obtained from Public Health Seattle King County and geocoded. Exposure was conceptualized as contact between stressors (FFRs) and receptors (participants’ mobility records from GPS data) using four proximities: 21 m, 100 m, 500 m, and ½ mile. Measures included count of proximal FFRs, time duration in proximity to ≥1 FFR, and time duration in proximity to FFRs weighted by FFR counts. Self-reported exposures (FFR visits) were excluded from these measures. Logistic regressions tested associations between one or more reported FFR visits and the three exposure measures at the four proximities. Time spent in proximity to an FFR was associated with significantly higher odds of FFR visits at all proximities. Weighted duration also showed positive associations with FFR visits at 21-m and 100-m proximities. FFR counts were not associated with FFR visits. Duration of exposure helps measure the relationship between the food environment, mobility patterns, and health behaviors. The stronger associations between exposure and outcome found at closer proximities (<100 m) need further research.

## 1. Introduction

Exposure to the food environment may influence both diet quality and obesity rates [1,2,3]. Living near fast food restaurants (FFRs) and far from supermarkets has been linked to low diet quality and high body mass index (BMI), especially in lower-income populations [4,5,6,7,8,9,10]. Such findings have sparked interest in restrictions on where local governments allow FFRs to be located [11,12].

However, much of the evidence underlying potential FFR restrictions has been derived from studies whose measures of exposure are counts of FFRs within some distance of a study participant’s home address [13,14]. These measures only capture food availability and accessibility near home, and ignore people’s daily mobility patterns, which frequently extend far beyond the home neighborhood and may or may not entail being near a FFR [15,16,17,18,19]. As a result, researchers have repeatedly called for more nuanced exposure measures that capture the complex and dynamic spatial experiences of modern urban life [20,21,22,23,24,25,26].

One opportunity to develop more nuanced measures draws on exposure science, which conceptualizes exposure as the result of contact between a stressor (a factor or environmental trait) and a receptor (an organism) [27]. Crucially, exposure science emphasizes that both stressor and receptor traits determine whether contact occurs. Measuring exposure to the food environment benefits from exposure science frameworks that examine the relationship between receptors (i.e., study participants) and stressors (e.g., FFRs) based on the receptor’s actual mobility patterns [28]. Mobility patterns are in turn affected by a range of receptor traits or characteristics (e.g., gender, race, income, food preferences).

Spatially explicit mobility data from global positioning systems (GPS) present an excellent opportunity for analyzing contacts between stressors and receptors, because GPS devices objectively record participants’ locations throughout the day [23,25]. Although theory and measurement of built environment exposure using GPS frameworks is an area of active research [20], there is a need for more detailed theoretical explanations.

Despite their limitations [29,30,31], GPS methods hold significant advantages over other methods (most notably participant recall) in identifying a person’s location in time (i.e., measure time spent in proximity to FFRs). Such time-based environmental exposure measures are common in the study of air pollution [32,33,34], but to our knowledge, only two have been used in studies of the food environment [35,36]. Cetateanu et al. [36] and Salder et al. [35] found positive associations between food purchasing behaviors and time spent within 50 m of food stores. In these studies, the goal was to measure adolescent and young adult food choices. Therefore, the studies excluded time spent in motorized vehicles based on the assumption that young people would not be able to make food choices independently of parental supervision when traveling in their parents’ cars. However, when considering adults, car travel may be yet another form of exposure to the built environment. Indeed, one of the potential strengths of time-based exposure may be that it captures differences in exposure that occur at driving speeds (e.g., 50 km/h) and at walking speeds (e.g., 5 km/h).

Most exposure measures created solely from GPS, including time-based measures, are subject to selective mobility bias [37]. This bias arises when purposeful exposure to stressors (e.g., encountering an FFR because of a planned visit to that FFR) is not excluded, because a subset of exposures (i.e., proximity to an FFR) will perfectly predict the outcome (i.e., visiting an FFR); that is, the exposure was itself the outcome.

This study introduces time-based objective measures of food environment exposure using GPS records and time-matched travel logs. We compared these measures to the FFR counts used in most previous studies [21,22]. In contrast to the two studies that use similar time-based measures, this study leverages travel diaries to account for the selective mobility bias [35,36]. We further detail a process by which GPS-based time durations inside buffered areas can be measured and compare the performance of different proximity buffer sizes in the prediction of the odds of visiting an FFR.

## 2. Materials and Methods

Data came from the Seattle Obesity Study II (SOSII), which employed a stratified, address-based sampling scheme to ensure spatial and economic variation. All of the participants were recruited from within King County, Washington (WA), which is an urbanized area with a population of two million within 450 mi^2^. Sampling, recruitment, and data collection have been described previously [38]. Briefly, eligible participants were aged 18 to 55 years with no mobility issues and who identified themselves as the primary food shoppers for their households. Verbal consent was obtained from 712 potential participants, of which 516 (72.5%) were enrolled in the study. Through in-person meetings, researchers obtained written consent, administered a computer-aided survey, and trained participants on how to fill out a seven-day place-based travel log modified from the National Household Travel Survey [39] and how to wear and recharge the GPS receiver (Qstarz BT-Q1000XT; Qstarz International Co., Ltd.; Taipei, Taiwan). In the travel log, participants were asked to record all of the places visited during each observation day upon time of arrival and the time of departure from each place. The GPS receiver was worn during the same seven-day period covered in the travel log. Participant data were collected between December 2011 and October 2012. All of the procedures were approved by the University of Washington Institutional Review Board (approval number 39737).

### 2.1. Geographic Information Systems Data

Participants’ residential and workplace addresses were geocoded using King County address point data [40] with ArcMap 10.2 (ESRI, Redlands, CA, USA). Of the 516 eligible residential addresses, 481 (93%) were matched automatically with a minimum match score of 100. The remaining were geocoded by manually referencing Google Maps. Of the 370 workplace addresses, 232 (63%) were matched automatically, and the remaining addresses were likewise manually geocoded. Of the 146 participants with no reported work place address, 104 (71%) were not employed outside the home.

Addresses for the 573 FFRs in King County came from Public Health Seattle King County 2012 food permit records, and were geocoded using the techniques described above. Venues were considered FFRs if they were part of national or regional chains that lack table service and sell inexpensive food in a short time span [41].

### 2.2. Dependent Variable

Places that participants reported visiting in the travel log were identified as FFRs if they met the fast food restaurant criteria [41]. Travel log entries showed that 36% of the sample reported visiting at least one FFR during the observation period. Participants were dichotomized to those with zero visits vs. one or more FFR visit.

### 2.3. Covariates

Several variables previously associated with fast food consumption were available for analysis: age (≤45; >45), gender, race/ethnicity (non-Hispanic white; other), household size (≤2; >2 persons in household), number of cars (≤1; >1), education (less than college degree; college degree or higher), and annual household income (<$50K; $50–100K; ≥$100K). The age variable was dichotomized due to a sample that skewed older as a result of our sampling goal targeting households’ primary food shoppers. Since 84% (*n* = 430) of the study sample was non-Hispanic white, all other race and ethnicity responses were collapsed into one category. Household size (≤2; >2 persons in household) was included because we hypothesized that larger households, especially those with children, would eat more fast food. Participants with missing responses (*n* = 18) to any of the questions or with responses of “Don’t know/not sure” were excluded from the sample.

The Euclidean distance between home and work was computed using ST_Distance in PostGIS (The PostGIS Development Group, PostGIS, 2008). Commute distance was categorized as: no commute; <8.4 km (sample median); ≥8.4 km [42]. Property value was used as a proxy for wealth [43,44,45], and residential density was used to account for neighborhood differences in travel distance and speed. Both measures leveraged the King County tax assessor parcel data [46] with ArcMap 10.1 (ESRI, Redlands, CA) to identify residential units within an 800-m radius of participants’ homes. Residential density was dichotomized at the sample median value of 1892 residences in the 800-m buffer. Property value was calculated as the average value of residential units located on the participants’ home parcel and split into tertiles: $38–227K; $227–323K; and ≥$323K (USD).

### 2.4. Exposure Measures

Each GPS point included latitude and longitude coordinates measured at configured intervals of 30 s (s) or less. Erroneous GPS points were removed following the procedures described by Tsui and Shalaby, and the Personal Activity and Location Measurement System [29,47,48]. The remaining points were linked in a temporal sequence to form line segments between each successive pair of points. Then, these line segments were used to estimate participant locations between GPS measurements. Each segment’s time interval was calculated as the difference in timestamps between the segment’s starting and ending points. Thus, the line segments enabled the delineation of each participant’s travel path, providing an individual-level measure of continuous physical and temporal exposure.

Segments with durations longer than 30 s accounted for only 0.14% of all segments in the sample and were excluded from the analyses because they were assumed to be the result of blocked satellite signals, or powered-off or malfunctioning data loggers.

Line segments crossing the county boundary were truncated at that boundary, thus limiting the observation of exposure to within King County. Since participants were inconsistent in following the instructions to turn off their GPS data loggers while they slept, GPS tracks measured at their homes were excluded from the study. Following the precedent set by Hurvitz et al. [49], line segments within a 125-m radial buffer of participants’ homes were excluded, and those that crossed into the buffer were truncated. Likewise, line segments within 125 m of workplaces [49] were excluded, because some participants turned off their data loggers while at work, and in other instances, satellite reception was hindered due to workplaces located underground or in or near tall buildings. The time intervals of truncated line segments were estimated by comparing the length-to-time ratio of the original segment to the length of the truncated segment. Data were processed using PostGIS 2.1 within PostgreSQL 9.19 (The PostgreSQL Global Development Group, 2008) in an R 3.2.1 programming environment (R Core Team, 2015, R Foundation for Statistical Computing, Vienna, Austria).

Proximity to FFRs was measured with four radial buffers around each FFR parcel: 21 m, 100 m, 500 m, and ½ mile. Early studies examining daily paths and GPS data [21,22,23,25] adopted these larger buffer sizes, which were used in even earlier studies exploring built environment characteristics near participants’ homes. Whereas the larger buffers were selected because they were used in prior studies, the 21-m buffer is new to built environment (BE) research. It represents the approximate width of many urban streets platted west of the Ohio River and is the maximum distance at which a human face can still be identified by another human [50]. With theory behind buffer size lacking [15], we posit that different buffer sizes capture different relationships between stressor and receptor. On the one hand, closer proximities may be capturing the receptor’s sensory experience of the stressor; what can be seen, heard, smelled, or even touched. On the other hand, farther proximities may require cognitive or other processes internal to the receptor that link the receptor to the stressor. Such processes may include knowledge or memory of stressor locations and attitudes or preferences for certain types of stressors [51].

The FFR count was the number of unique buffers through which a participant’s GPS data intersected over the course of each observation day. Duration of exposure was calculated as the total minutes per day spent in proximity to one or more FFRs. It was estimated by summation of the time intervals of complete and truncated GPS line segments intersecting the areas within FFR proximity buffers for each observation day. Weighted duration was calculated as the duration of exposure weighted by the number of FFRs in proximity to the participant. It was estimated by summation of the line segment intervals that intersected with each individual FFR buffer, regardless of whether buffers overlapped or not. This multiplied the time value of line segments inside overlapping proximity buffers by the number of buffers. Count, duration, and weighted duration exposure measures were averaged across each participant’s observation days to create daily averages per participant. Due to zoning regulations and variations in building density, some areas of the county have very high concentrations of FFRs, while other areas are devoid of FFRs. Therefore, we expected the distribution of these variables to be highly skewed, especially for the larger proximity buffer sizes. For this reason, as well as for ease of comparison across proximities, exposure measures were converted into tertiles.

To minimize selective mobility bias, GPS data corresponding to FFR visits reported in the travel diary were removed from the count, as well as the duration of exposure measures. This was accomplished by excluding GPS line segments that intersected the FFR proximity buffers within ± 10 min (to account for recall bias) of the FFR visits reported in the travel logs. Failure to exclude these visits would have resulted in attributes of FFR visitation estimating both predictor and outcome.

Duration of exposure was estimated by summation of the intervals between GPS points (Figure 1). Where line segments crossed the buffer, the time interval of the portion of the segment inside the buffer was estimated using the time-to-length ratio of the complete line segment. The length of the partial segment inside the buffer (e.g., 17.4 m), the complete interval time (e.g., 10 s), and the complete length of the segment measured in GIS (e.g., 26.7 m) were used to estimate the time interval inside the buffer (10 s × 17.4 m/26.7 m = 6.5 s). When the buffers of two or more FFRs overlapped, the duration of exposure was computed as the total time spent in one or more FFR buffers (e.g., 9.3 min + 8.5 min + 3.0 min = 20.8 min). The weighted duration weights the amount of time spent in multiple buffers by the number of overlapping buffers (e.g., 9.3 min + (2 buffers × 8.5 min) + 3.0 min = 29.3 min).

When line segments intersected proximity buffers around FFRs, the time-to-length ratio of the complete segment was used to estimate the time of the portion of the segment inside the buffer based on its length. Truncated segment time equals complete segment time multiplied by the ratio of truncated segment length to complete segment length. When the buffers of two or more FFRs overlapped, the duration was computed as the total time spent in one or more FFR buffers (9.3 min + 8.5 min + 3.0 min = 20.8 min). The weighted duration weighted the amount of time spent in multiple buffers by the number of overlapping buffers (9.3 min + (2 buffers × 8.5 min) + 3.0 min = 29.3 min).

### 2.5. Analysis

The sociodemographic distribution of the sample by FFR visitation and the three exposure variables was examined using proportions and chi-squared tests. Means (SD) of the exposure measures at the four proximities (21 m, 100 m, 500 m, and ½ mile) were calculated. Bivariate associations of FFR visitation with tertiles of the exposure variables were examined using chi-squared tests.

Multivariate logistic regressions with robust standard errors examined the odds of visiting an FFR by each exposure measure tertile at each proximity buffer. Twelve models compared differences in the three exposure measures at the four different proximity sizes, and included sociodemographic variables, commute distance, and residential density as covariates. Analyses were conducted using R package version 3.2.1. The glm function (family = ‘binomial’) was used to calculate logistic regressions. Robust standard errors were manually calculated using R.

## 3. Results

Of the 516 participants, 477 had concurrent GPS and travel log data. Sixteen of these were removed due to an insufficient number of observation days (<3 days) or recording errors in either GPS data or their travel log. Another 18 participants were removed due to missing survey responses. Eight participants with workplaces outside King County were removed due to the possibility of daily unmeasured FFR exposures occurring outside the county. Another 23 participants did not provide any work addresses. The final analytical sample consisted of 412 participants (see Figure 2). The mean number of days that subjects reported in the travel log was 6.6 (SD 0.9). There were 276 FFR visits reported in the travel logs by 149 (36.2%) participants. We verified 182 (71.7%) of the 254 reported FFR visits in King County by identifying GPS data showing that the participant was in proximity to an FFR during the visit times reported in the travel log. About 58% (*n* = 81) of FFR visitors had all of their visits verified by GPS, while 23% (*n* = 32) participants did not have any of their visits recorded by GPS.

The sample was disproportionately over age 45 (62%); female (71%); non-Hispanic white (79%); and college graduates (62%). The sample was evenly distributed across three categories of income (28.6% < $50K, 36% $50–100K, and 35% ≥ 100K). Most of the sample was married (67%); half of the sample lived in households with three or more persons; and 63% had >2 cars in the household (Table 1).

Significant sociodemographic differences (*p* < 0.05) were observed between participants reporting FFR visits and those who did not. FFR visitors were less likely to have a college degree (53.0% vs. 66.9%), and more likely to live in households with more than two people (58.4% vs. 44.5%) and with at least two cars (70.5% vs. 58.6%). Visitors to FFRs were also more likely to have commutes longer than the sample median (41.6% vs. 28.5%) and live in neighborhoods below the sample median residential density (63.8% vs. 42.2%).

Table 2 displays the mean level of exposure to FFRs at each proximity level. Participants spent on average one minute a day (SD 1.8) within 21 m of at least one FFR, and were within 21 m of 1.5 FFRs (SD 1.1) per day on average. This results in a mean daily weighted duration of 1 min (1.9 SD) at this proximity. At the ½-mile proximity, participants averaged 117.7 min (SD 69.2) in proximity to an average of 34.1 FFRs (SD 18.9) per day, or a mean weighed duration of 607.6 min (SD 526.9).

Table 3 presents bivariate associations between tertiled exposure variables and FFR visits. Visiting an FFR was not associated with the count of FFRs to which participants were exposed at any measured proximities. However, both duration and weighted duration were significantly associated (*p* < 0.05) with an increased likelihood of visiting an FFR at the 21-m and 100-m proximities.

Table 4 presents associations between FFR exposure and FFR visits after accounting for a priori identified key sociodemographic covariates (age, gender, race/ethnicity, education, income, household size, and residential property value) and commute patterns (number of cars in household, commute distance, and residential density). There were no significant associations between FFR count exposure and FFR visits at any proximity. In contrast, the odds of reporting one or more FFR visits was positively associated with the duration of exposure at all four proximities. A positive relationship was observed between tertiles of duration and probability of FFR visits for a 21-m proximity buffer (β: 2.06; 95% CI: 1.17–3.65 for tertile 2, and β: 2.80; 95% CI: 1.58–4.96 for tertile 3, with tertile 1 as the reference) and ½-mile buffer (β: 1.93 and 2.96, respectively). For the 100-m and 500-m proximity buffers, only tertile 3 showed significant positive associations between the duration of exposure and FFR visits (β: 2.89; 95% CI: 1.65–5.07 for 100 m, and β: 1.72; 95% CI: 1.20–9.4 for 500 m). Weighted duration was significantly associated with increased odds of FFR visits at only the 21-m and 100-m proximities (OR 2.69, 95% CI 1.53–4.73, and OR, 3.07, 95% CI 1.76–5.36, respectively).

## 4. Discussion

Quantifying how long a person is exposed to the food environment that lies in his or her path may be useful in evaluating associations between the food environment and behavior. Using FFRs as stressors, this study showed that the amount of time spent daily near FFRs predicted FFR visits, but the number of FFRs along a person’s daily path did not. Weighting the duration of exposure by the number of FFRs only predicted an FFR visit at the 21-m and 100-m proximities, further suggesting that the counts were an ineffective measure of exposure. Indeed, in these close proximities, simple and weighted duration measures had similar value ranges, because smaller buffers are less likely to overlap with each other. Overall, the duration of exposure in closer proximity to a stressor better captured the relationship between environment and behavior than the count of stressors at any proximity.

It is difficult to explain why FFR count was not significant in this study compared to other studies [21,22] given the differences in the studies’ populations, locations, and related built environment. Additionally, differences included the studies’ dominant travel mode and the use of simulated motorized commute trips, rather than the GPS-based data used in the current study, which included any travel mode. In general, trips taken on foot are shorter than those taken by automobile or mass transit. Therefore, walkers can be expected to come into the proximity of fewer FFRs, although the influence of the FFRs may be stronger as walkers will have more direct sensory input than people using motorized travel modes.

The weaker associations found between exposure and outcome at the farther proximities (500 m and ½ a mile) raise important questions for future exposure research. Although larger buffers may work for capturing environmental exposures near participants’ homes [52,53,54], our findings suggest that they are less efficacious in measuring exposure along people’s actual daily path. Research on buffer size is limited, and there are very few theoretical explanations in the literature justifying those sizes [15].

In interpreting our findings, we rely on speculation and personal judgment. First, as the buffer size increases, the level of spatial variation between participants decreases. This alone may account for our findings. However, there may also be a saturation effect in which exposure duration may reach a level where longer contact does not lead to a different behavior. Exposure saturation may be explained by the number of stressors naturally increasing with the size of the area considered for possible contact: if a participant were placed at a random location in King County, he or she would have a 0.07%, 1.3%, 16.3%, and 32.2% chance of being exposed to an FFR at 21 m, 100 m, 500 m, and ½ mile distances, respectively. In this study, on an average day, participants found themselves near more than four times the number of FFRs in the ½-mile buffer than in the 100-m buffer. The study also pointed to plateaus of exposure duration, as spending more than 190 min a day within 100 m of an FFR (tertile 3 OR 2.89 95% CI 1.65–5.07) was not associated with the higher risk of an FFR visit. Overall, participants spending an average of 84.4 min per day or more within 500 m to ½ mile of an FFR were not more likely to visit a FFR than those who spent an average of 17 min or less within 100 m or 21 m of an FFR.

In addition, the higher strength of association between duration of exposure and outcome in the closer proximities to stressors brings into question results from past studies, which focused on farther proximities. With few exceptions, past studies of food environment exposure only considered participants’ home or work locations, and not actual, dynamic locations over the course of daily life. They explored exposure at greater proximities (>500 m) ostensibly because they aimed to capture the neighborhood that participants might “know”, the neighborhood that might be walked, or neighborhood destinations that could be accessed in a reasonable amount of time [17,52,53,54]. The resulting exposure measures may have reflected individuals’ hypothesized cognitive aspects of neighborhood and possible access to specific environments. Of note, one the earliest GPS-based studies of proximity to FFRs and health behaviors also used a ½-mile proximity buffer [23].

In contrast, this study, with its emphasis on mobility patterns, was able to consider both closer and farther proximities. It showed that when using buffer radii ranging from 21 m (equivalent to a 16-s walk) to half a mile (a 10-min walk), shorter (less than one-minute walk) distances between stressors and receptors were able to better capture associations between environment and behavior. The closer proximities delineated a participant’s immediate sensory range, which may have been linked to their behavior.

Further research is needed in order to explore the relationship between a person’s exposure to the environment, his or her sensory experience, and his or her behavior. It needs to determine the size of the buffer that best captures sensory experience. The 21-m proximity can catch street-level experiences, and the 100-m proximity, which is approximately the length of a city block, encompasses the block level. Even within close proximities, sensory experiences can vary greatly by location and speed of travel. In dense urban settings, visual range may be framed by street trees, buildings, and other structures while in less dense suburban settings, the streets are wider and the buildings are farther apart [55,56]. The narrow visual range caters to slower travel speeds, while wider visual ranges can facilitate higher travel speeds. As such, the wider visual ranges are one of the many factors that make suburban areas more automobile-friendly [55,57]. Thus, a time-based measure of exposure decreases the need to gauge the effect of different proximities based on travel mode [21,22].

Broadly, considering environmental exposure in research using mobility patterns demands new measurement paradigms [58]. Chaix et al. previously identified selective mobility bias (the self-selection of places as well as paths or routes) as demanding analyses that distinguish between willful and haphazard, or active and passive exposure to environment [37]. The present work goes further and highlights the need to identify appropriate proximities or exposure buffer sizes to capture visual/audio sensory contact between stressors and receptors. The results strongly suggest that immediate visual/audio sensory contact may impact the behavior, whereas the cognitive and accessibility aspects of the FFRs within the neighborhood appear to be less relevant.

Future researchers ought to use spatially and temporally precise measures of exposure to the food environment in order to consolidate the theoretical constructs underlying the effect that the foodscape has on behavior. It will also need to examine differences between outlet types, as well as possible interactions between the broader urban and suburban contexts, seasonal variations, and related travel patterns [14,54,59,60].

Whereas all aspects of exposure (stressors, receptors, and proximities) were measured objectively, FFR visitation came from self-reported data. We were able to GPS verify 71.7% of reported visits, but 32 participants had no reported visits verified. Unverified visits represent instances in which the GPS data indicate that a participant was not at the location reported in the travel log during the reported arrival and departure times. The most likely causes of unverified visits were: a) participant error in reporting arrival and departure times, and b) participants failing to carry the GPS logger during their FFR visit. When the 32 participants were removed from analysis, the odds for each of the exposure measures retained their levels of significance.

Conversely, participants may have visited FFRs without recording those visits in the travel log (due to recall or social desirability bias). The use of wearable cameras [61,62] or the use of space–time clustering algorithms such as ST_DBSCAN [63] could aid in detecting unreported FFR visits. The visual inspection of GPS tracks or processing algorithms might be used to detect instances of unreported FFR visitation. However, GPS tracks in and of themselves may not be precise enough to locate with certainty someone being within the relatively small spaces of FFRs. Schipperijn et al. found that the median error between actual locations and GPS-derived locations using the same model GPS data logger employed in this study was 2.9 m, but this value may increase to 10 m depending on the conditions of measurement [30]. Also, it may be difficult, if not impossible to distinguish between an FFR visit and an individual passing by or loitering near an FFR, when taking into account that a 3-min FFR visit captured with a 30-s GPS measurement interval may be represented by six or fewer data points, and the average speed of service for a McDonalds drive-through is only 189.5 s [64]. Moreover, spatial variation in GPS error (e.g., due to tall buildings blocking GPS signals) might result in a spatially differential error regarding the probability of identifying a true FFR visit [30,65].

## 5. Conclusions

This study offers a new approach to measure exposure to the food environment over the course of daily life, incorporating GPS-based participants’ location and travel log data to quantify realized exposure rather than simple access from home or work. Duration of exposure was measured precisely by associating the temporal intervals of GPS points with their corresponding spatial geometries.

Time-based exposure to the food environment seemed to better capture its potential influence on health than counts or densities of food outlets. Time spent within 100 m or less of a FFR was strongly associated with the odds of visiting an FFR. Further research on environmental exposure to FFRs should consider the duration of exposure based on mobility patterns and take into account sensory and cognitive influences on behavior.

## Figures and Tables

**Figure 1 ijerph-16-01180-f001:**
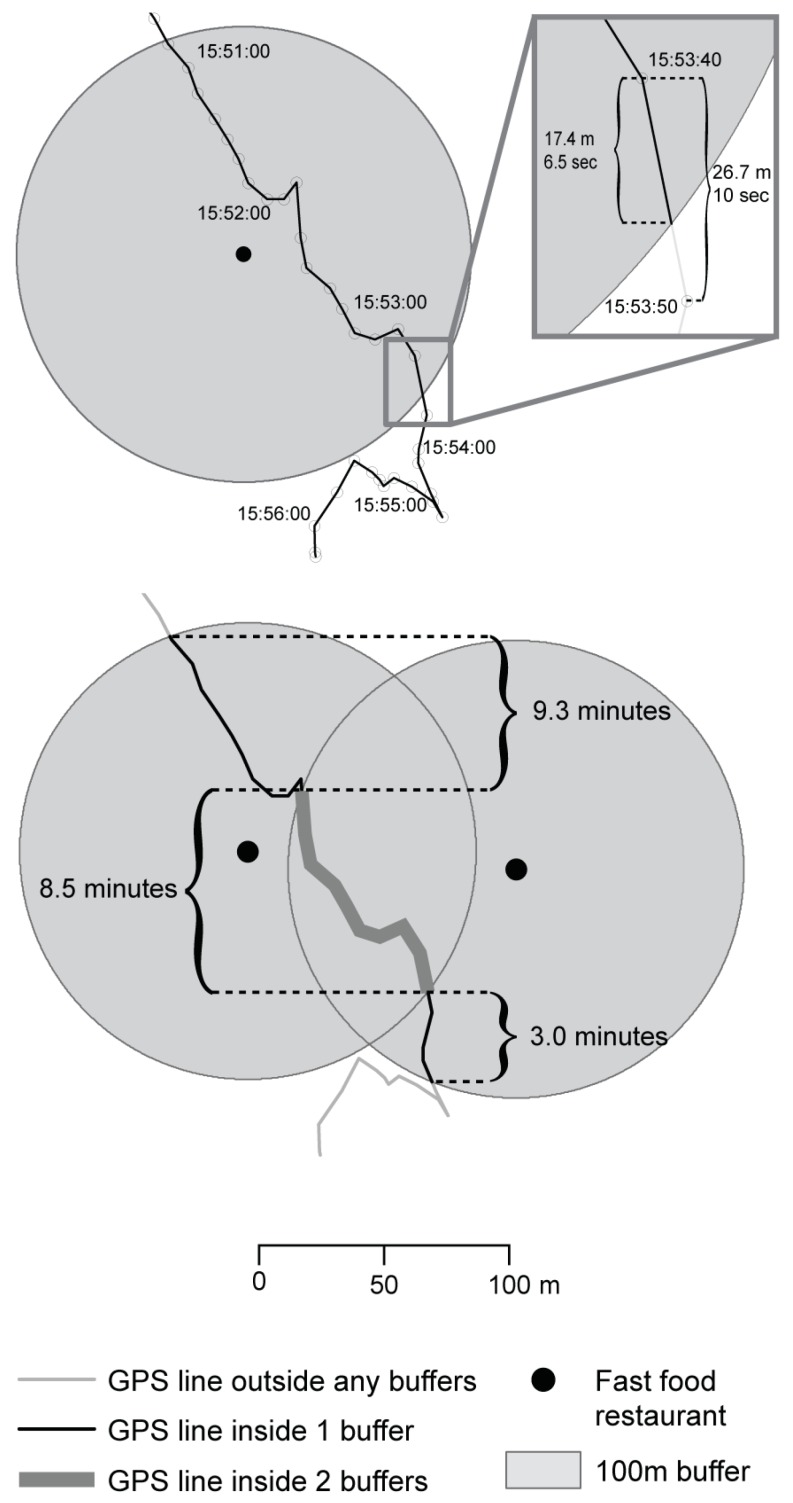
Estimation of time spent in proximity to fast food restaurants (FFRs).

**Figure 2 ijerph-16-01180-f002:**
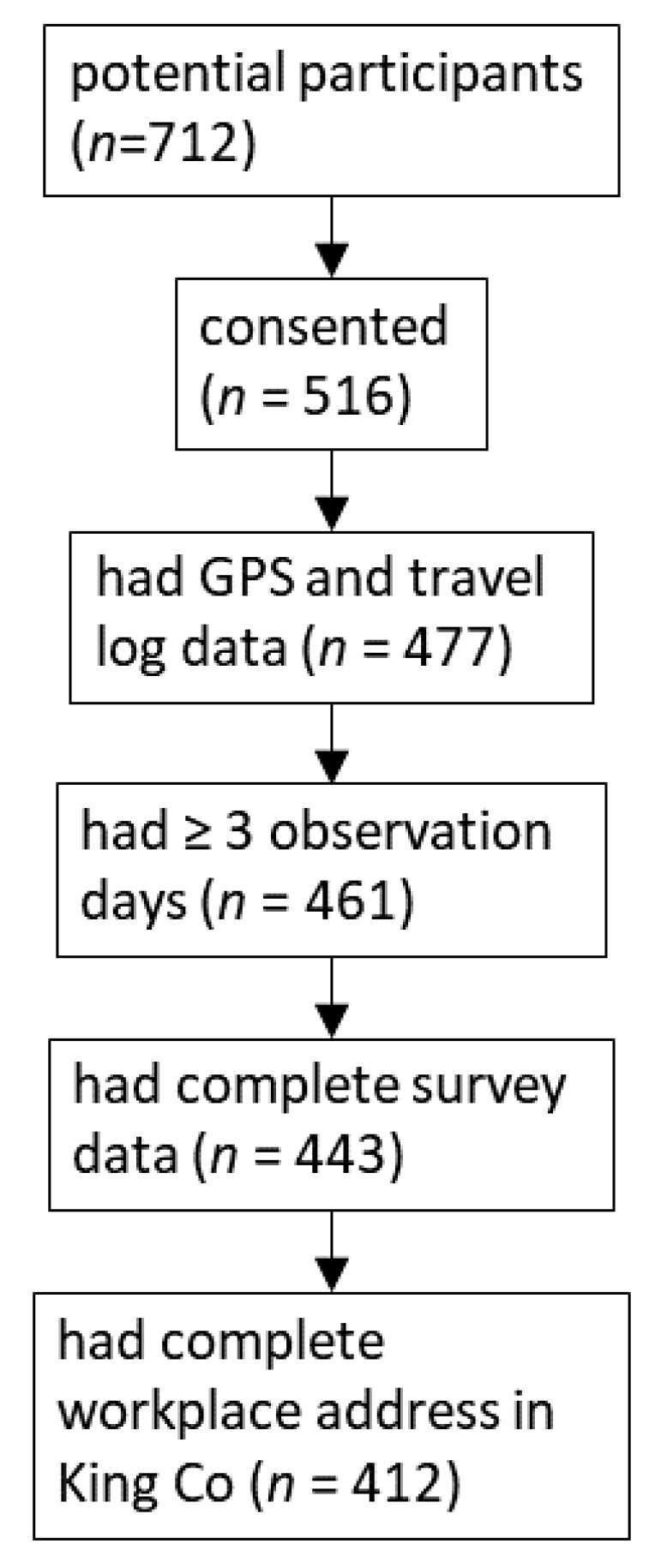
Subject inclusion criteria and sample size.

**Table 1 ijerph-16-01180-t001:** Descriptive characteristics of FFR (fast food restaurant) visitors and non-visitors.

Total	*N*	No Reported Visits	One or More Reported Visits	*p*-Value ^1^
		*n* (%)	*n* (%)	
	412	263 (100)	149 (100)	
Age (years)				0.432
<45	157	96 (36.5)	61 (40.9)	
≥45	255	167 (63.5)	88 (59.1)	
Gender				0.728
Female	293	185 (70.3)	108 (72.5)	
Male	119	78 (29.7)	41 (27.5)	
Race				0.999
White non-Hispanic	327	209 (79.5)	118 (79.2)	
Non-White	85	54 (20.5)	31 (20.8)	
Education				0.007
Some college or less	157	87 (33.1)	70 (47.0)	
College graduate	255	176 (66.9)	79 (53.0)	
Income				0.874
<$50K	118	76 (28.9)	42 (28.2)	
$50–100K	151	94 (35.7)	57 (38.3)	
≥$100K	143	93 (35.4)	50 (33.6)	
Household size				0.044
1–2	200	138 (52.5)	62 (41.6)	
≥3	212	125 (47.5)	87 (58.4)	
Property value				0.704
$38–227K	136	90 (34.2)	46 (30.9)	
$227–323K	137	84 (31.9)	53 (35.6)	
≥$323K	139	89 (33.8)	50 (33.6)	
Number of cars in HH				0.022
≤1	153	109 (41.4)	44 (29.5)	
≥2	259	154 (58.6)	105 (70.5)	
Commute distance				0.005
No commute	138	87 (33.1)	51 (34.2)	
<Median (8.4 km)	137	101 (38.4)	36 (24.2)	
>Median (8.4 km)	137	75 (28.5)	62 (41.6)	
Residential density				0.001
<Median density (1892 residences)	206	111 (42.2)	95 (63.8)	
>Median density (1892 residences)	206	152 (57.8)	54 (36.2)	

^1^ Derived from chi-squared analysis. HH: household.

**Table 2 ijerph-16-01180-t002:** Means (SD) of daily FFR exposure by proximity buffer.

Exposure	Buffer Distance			
	21 m Mean (SD)	100 m Mean (SD)	500 m Mean (SD)	½ mile Mean (SD)
Count of FFRs in buffer per day	1.5 (1.1)	8.1 (4.5)	24.34 (13.2)	34.1 (18.9)
Duration of exposure ^1^	1.0 (1.8)	17.0 (16.6)	84.8 (56.7)	117.7 (69.2)
Weighted duration ^1^	1.0 (1.9)	22.7 (22.0)	297.1 (247.4)	607.6 (526.9)

^1^ In minutes per day.

**Table 3 ijerph-16-01180-t003:** Descriptive characteristics of FFR visitors and non-visitors by FFR exposure measures.

Buffer Distance, Tertiles of Exposure	*N*	No Reported Visits (*n*)	One or More Reported Visits (*n*)	*p*-Value ^1^
FFR count ^2^				
21 m				0.934
0–0.86	123	80 (30.4)	43 (28.9)	
0.86–1.71	140	88 (33.5)	52 (34.9)	
1.71–8.00	149	95 (36.1)	54 (36.2)	
100 m				0.076
0–5.82	136	95 (36.1)	41 (27.5)	
5.82–9.14	137	89 (33.8)	48 (32.2)	
9.14–27.2	139	79 (30.0)	60 (40.3)	
500 m				0.380
0–17.00	139	95 (36.1)	44 (29.5)	
17.00–28.40	133	83 (31.6)	50 (33.6)	
28.40–78.60	140	85 (32.3)	55 (36.9)	
1/2 mile				0.385
1 to 23.00	138	91 (34.6)	47 (31.5)	
23.00–40.50	134	89 (33.8)	45 (30.2)	
40.50–115.00	140	83 (31.6)	57 (38.3)	
Duration of exposure ^3^				
21 m				0.009
00:00:00–00:00:09	136	99 (37.6)	37 (24.8)	
00:00:09–00:00:39	136	87 (33.1)	49 (32.9)	
00:00:39–00:12:54	140	77 (29.3)	63 (42.3)	
100 m				0.001
00:00:00–00:08:58	136	100 (38.0)	36 (24.2)	
00:08:58–00:17:06	136	91 (34.6)	45 (30.2)	
00:17:06–03:10:00	140	72 (27.4)	68 (45.6)	
500 m				0.188
00:00:00–00:57:06	136	92 (35.0)	44 (29.5)	
00:57:06–00:01:32	136	90 (34.2)	46 (30.9)	
00:01:32–08:20:00	140	81 (30.8)	59 (39.6)	
1/2 mile				0.085
00:06:59–01:21:00	136	97 (36.9)	39 (26.2)	
01:21:00–02:08:00	136	82 (31.2)	54 (36.2)	
02:08:00–09:05:00	140	84 (31.9)	56 (37.6)	
Weighted duration ^3^				
21 m				0.006
00:00:00–00:00:09	136	97 (36.9)	39 (26.2)	
00:00:09–00:00:41	136	91 (34.6)	45 (30.2)	
00:00:41–00:12:54	140	75 (28.5)	65 (43.6)	
100 m				0.001
00:00:00–00:11:24	136	101 (38.4)	35 (23.5)	
00:11:24–00:23.06	136	89 (33.8)	47 (31.5)	
00:23:06–03:14:00	140	73 (27.8)	67 (45.0)	
500 m				0.290
00:00:00–02:59:00	136	93 (35.4)	43 (28.9)	
02:59:00–05:02:00	136	87 (33.1)	49 (32.9)	
05:02:00–32:00:00	140	83 (31.6)	57 (38.3)	
½ mile				0.424
00:06:59–05:49:00	136	91 (34.6)	45 (30.2)	
05:49:00–10:26:00	136	81 (30.8)	55 (36.9)	
10:26:00–73:40:00	140	91 (34.6)	49 (32.9)	

^1^ Derived from chi-squared analysis; ^2^ Counts of fast food restaurants per day within buffer; ^3^ In hh:mm:ss format.

**Table 4 ijerph-16-01180-t004:** Logistic regression using robust standard errors to predict FFR visitation.^1^

Exposure	21 m		100 m		500 m		Half Mile	
	Odds Ratio	95% CI	Odds Ratio	95% CI	Odds Ratio	95% CI	Odds Ratio	95% CI
FFR count								
Tertile 1	Ref		Ref		Ref		Ref	
Tertile 2	1.26	0.73–2.18	1.16	0.66–2.04	1.32	0.76–2.3	1.06	0.6–1.86
Tertile 3	1.41	0.8–2.47	1.68	0.96–2.93	1.38	0.76–2.51	1.49	0.83–2.68
Duration								
Tertile 1	Ref		Ref		Ref		Ref	
Tertile 2	2.06 *	1.17–3.65	1.24	0.7–2.18	1.06	0.61–1.83	1.93 *	1.1–3.39
Tertile 3	2.8 ***	1.58–4.96	2.89 ***	1.65–5.07	1.72 *	1–2.94	2.16 **	1.22–3.83
Weighted duration								
Tertile 1	Ref		Ref		Ref		Ref	
Tertile 2	1.62	0.92–2.85	1.4	0.79–2.47	1.15	0.67–1.99	1.25	0.72–2.17
Tertile 3	2.69 **	1.53–4.73	3.07 ***	1.76–5.36	1.47	0.86–2.52	1.15	0.67–1.99

^1^ Adjusted for age, gender, race, education, income, number of cars in household, household size, commute distance, and residential density. * *p* < 0.05; ** *p* < 0.01; *** *p* < 0.001.

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
