# Peer review of "A Time-Based Objective Measure of Exposure to the Food Environment"

_ijerph, 2019, doi:10.3390/ijerph16071180_

Round 1
Reviewer 1 Report
Thank you for inviting me to review at IJERPH. This is a nice paper that undertakes an interesting and important analysis. It is written well throughout and was a pleasure to read. Indeed, it is refreshing to review a paper that clearly a lot of work has been put into and one that I have few suggestions for. The authors should be proud of a good output, well done.
The introduction is a little pedestrian, but covers the main areas and it is clear the importance/purpose of the paper. Methods used appear appropriate and the data/analyses are novel and help answer the research gaps identified. The topic is interesting to the journal readership and I can see become a useful reference for future study.
I have one issue. Four buffers are used - 21m, 100m, 500m and 1/2mile. The latter two do not seem to fit with the narrative of the paper. If one of your key arguments is that we need better data to understand how individuals are exposed in their environments, why then select two options with large buffer sizes that undermines the fact that you have their GPS location and therefore know their exact exposures. There might a good reason for this (I doubt it tbh), however it is never really made. While this might be overlooked, it is talked a lot about in the discussion:
"First, it appears that exposure duration may reach a level of saturation where longer contact does not lead to a different behavior." (lines 292-293)
You do not actually measure or test such saturation and your changing buffer sizes do not really capture it as you suggest. The larger buffer sizes really do not capture any exposure since an individual will not be aware of what is in these larger zones (the justification of immediate exposure is lost here). It also ends up reducing your spatial variation as the larger zones will produce more similar results between individuals, hence maybe why results disappear. A weak justification is given for the larger buffers:
"The 500 m and 1/2 mile proximities covered areas that participants could not immediately and entirely experience, but might be aware of, based on cognitive processes and memories. Exposure at these farther proximities did not appear to have an effect on behavior." (lines 319-321)
No evidence is given to support this assertion and I do not really agree with it at all. The discussion spends too much time talking about the different buffer zones that it becomes distracting and ignores the main point of their paper. I would refocus the text back to the novelty and importance of the paper set up at the start. That will make for a tighter paper and one that fairly reflects on its results. You have a good paper, don't lose sight!
The rest of my comments are very minor:
- Why keep age as a binary? And why split by 45 years? This is not clear and seems an odd decision.
- 'Airline distance' is cited but I would refer to this as euclidean distance. Seems odd that you would calculate that and not actual road network distance given all of the good spatial data work you do elsewhere - seems lazy.
- Line segments were truncated at the boundary of Kings County - does this introduce edge effects into your analysis?
- Table 1 includes '# of cars' rather than number of cars which is a little lazy
- "Table 46" (line 257) - I think this needs a dash in it
- Tables 4-6 are reported as ‘odds’ but in text as odds ratios - I suggest you be clear and consistent throughout
All in all a good paper, well done
Author Response
Reviewer comments are show in Courier 11 point, and our responses are shown in Calibri bold italic 11 point directly below each comment.
R | Thank you for inviting me to review at IJERPH. This is a nice paper that undertakes an interesting and important analysis. It is written well throughout and was a pleasure to read. Indeed, it is refreshing to review a paper that clearly a lot of work has been put into and one that I have few suggestions for. The authors should be proud of a good output, well done. |
Thank you for such kind words! | |
R
| The introduction is a little pedestrian, but covers the main areas and it is clear the importance/purpose of the paper. Methods used appear appropriate and the data/analyses are novel and help answer the research gaps identified. The topic is interesting to the journal readership and I can see become a useful reference for future study. |
Reviewer comments on earlier drafts of this paper have indicated that this subject matter is of interest to a wide range of disciplines. Finding the right wording and language so that our manuscript is of interest and relevant to that range has proven tricky. The end result is that our writing has indeed become pedestrian, but we hope that it makes our work more accessible. | |
R | I have one issue. Four buffers are used - 21m, 100m, 500m and 1/2mile. The latter two do not seem to fit with the narrative of the paper. If one of your key arguments is that we need better data to understand how individuals are exposed in their environments, why then select two options with large buffer sizes that undermines the fact that you have their GPS location and therefore know their exact exposures. There might a good reason for this (I doubt it tbh), however it is never really made. While this might be overlooked, it is talked a lot about in the discussion: |
In the present manuscript we note that the larger buffers come from earlier GPS food environment studies. Clarifying language was added to Section 2.4: Exposure Measures in the manuscript and in Section 4: Discussion. We do agree with the reviewer’s discomfort! The only reason for including the two larger buffers is that they have been used in many papers over the course of around 10 years of research on exposure to the BE. We think that the previous use of large proximities came from confusing access to certain features of the BE with exposure to the BE. Clearly people need to have access to places that support healthy behaviors (e.g. fresh and affordable fruit and vegetables), but also their behavior can be influenced negatively or positively by what is near them. Two researchers have made a clear difference between access and exposure, Basile Chaix and Mei-Po Kwan. Chaix measures exposure based on travel patterns and trips characteristics. Kwan measures exposure over the entire life course. For example, see
Chaix, B., 2018. Mobile Sensing in Environmental Health and Neighborhood Research. Annual Review of Public Health, 39, pp.367–384.
Perchoux, C. et al., 2014. Assessing patterns of spatial behavior in health studies: Their socio-demographic determinants and associations with transportation modes (the RECORD Cohort Study). Social science & medicine (1982), 119, pp.64–73. Available at: http://www.ncbi.nlm.nih.gov/pubmed/25150652.
Thierry, B., Chaix, B. & Kestens, Y., 2013. Detecting activity locations from raw GPS data: a novel kernel-based algorithm. International journal of health geographics, 12, p.14. Available at: http://www.pubmedcentral.nih.gov/articlerender.fcgi?artid=3637118&tool=pmcentrez&rendertype=abstract.
Kwan, M.-P., 2002. Feminist Visualization: Re-envisioning GIS as a Method in Feminist Geographic Research doi:10.1111/1467-8306.00309. Annals of the Association of American Geographers, 92(4), pp.645–661. Available at: http://www.blackwell-synergy.com/doi/abs/10.1111/1467-8306.00309.
Kwan, M.-P. & Lee, J., 2003. Geovisualization of Human Activity Patterns Using 3D GIS: A Time-Geographic Approach. In M. F. Goodchild & D. G. Janelle, eds. Oxford: Oxford University Press, pp. 48–66. | |
R | You do not actually measure or test such saturation and your changing buffer sizes do not really capture it as you suggest. The larger buffer sizes really do not capture any exposure since an individual will not be aware of what is in these larger zones (the justification of immediate exposure is lost here). It also ends up reducing your spatial variation as the larger zones will produce more similar results between individuals, hence maybe why results disappear. |
We modified Section 4: Discussion to take into account your comments. | |
R | A weak justification is given for the larger buffers: No evidence is given to support this assertion and I do not really agree with it at all. The discussion spends too much time talking about the different buffer zones that it becomes distracting and ignores the main point of their paper. I would refocus the text back to the novelty and importance of the paper set up at the start. That will make for a tighter paper and one that fairly reflects on its results. You have a good paper, don't lose sight! |
We agree that the paper would be much tighter if we left out the section on buffer sizes, however we think this links the paper to previous research and provides some much needed speculation on the processes that link proximity to exposure to behavior to health. If the reviewers disagree, we are more than happy to cut this section from the paper.
As for the quote, we have deleted it. | |
R | Why keep age as a binary? And why split by 45 years? This is not clear and seems an odd decision. |
The dichotomization of the age variable was due to a sample that skewed older as a result of our sampling goal targeting households’ primary food shoppers. | |
R | 'Airline distance' is cited but I would refer to this as euclidean distance. Seems odd that you would calculate that and not actual road network distance given all of the good spatial data work you do elsewhere - seems lazy. |
We’ve changed “airline” to “Euclidean.” We choose Euclidean buffers over network buffers so that we could link our work to the previous research on the topic. Our research is in part an elaboration on Burgoine & Monsivais (2013) and Burgoine et al. (2014). Both studies used 100- and 500-m Euclidean buffers as their measure of exposure. Further, Euclidean buffers allow for exposure during travel that is not occurring along a street network, such as cutting through parking lots, or green parks.
Burgoine, T., & Monsivais, P. (2013). Characterising food environment exposure at home, at work, and along commuting journeys using data on adults in the UK. The International Journal of Behavioral Nutrition and Physical Activity, 10(1), 85. https://doi.org/10.1186/1479-5868-10-85
Burgoine, T., Forouhi, N. G., Griffin, S. J., Wareham, N. J., & Monsivais, P. (2014). Associations between exposure to takeaway food outlets, takeaway food consumption, and body weight in Cambridgeshire, UK: population based, cross sectional study. BMJ (Clinical Research Ed.), 348, g1464. https://doi.org/10.1136/bmj.g1464 | |
R | Line segments were truncated at the boundary of Kings County - does this introduce edge effects into your analysis? |
In our study edge effects are concerning because they may result in underestimation of FFR exposure. Short of reviewing each of the line segments comprising the GPS we cannot discount edge effects, though we did take steps to minimize them. Namely, we removed participants from the analysis if their workplace was outside the county. We also reviewed all of the instances in which the GPS did not confirm a visit reported in reported in the travel diary. Edge effects were assumed to be at play in these situations, however that was not the case. | |
R | Table 1 includes '# of cars' rather than number of cars which is a little lazy |
Change made | |
R | "Table 46" (line 257) - I think this needs a dash in it |
Per comment R2.12, we've consolidated Tables 4 through 6 into one table. | |
R | Tables 4-6 are reported as ‘odds’ but in text as odds ratios - I suggest you be clear and consistent throughout |
Changes made | |
R | All in all a good paper, well done |
Again we greatly appreciate your kind words. |
Reviewer 2 Report
Thank you for the opportunity to review the manuscript entitled “A time-based objective measure of exposure to the food environment”. This paper used participant GPS data to use a new methodological measure of exposure to the food environment, duration. The paper demonstrates the need for a more nuanced approach to determining individual’s exposure to the food environment that extends beyond arbitrary boundaries using buffers around individual’s home and workplaces. Overall, I feel the paper makes a good contribution but there are aspects of the methods, given this is primarily a methodological paper, that need clarification and justification to highlight the utility of this approach. Further, I am not convinced on the relationship between the exposure (duration) and outcome (visits). These concerns are highlighted below in order of where they appear.
Abstract
Line 19- include the age of the participants.
Line 30- My major concern is that people who visit a fast food restaurant would have substantially higher amounts of time spent in proximity to FFRs, thus the exposure and outcome are the same?
Introduction
I think a larger discussion of advantages and limitations of travel logs is warranted in the introduction as your paper heavily relies on these. How many FFR visits do you feel were missed by relying on travel logs? Other studies have used individual-level GPS data to inform time-based measures of participants visits to destinations (Brusilovskiy et al., 2016) and children’s visits to food retails outlets (Chambers et al 2017 also used wearable cameras) using the ST DBSCAN algorithm (references provided at the bottom).
Methods:
Lines 94 & 134 – What was the data completeness over those seven days? How many Erroneous GPS points were removed, how much of a problem was this?
Given data loss is more likely when participants are free-moving, does the data fairly reflect participants normal activities?
Line 95 – Was there any effect of seasonality in participants’ mobility patterns or FFR visits? Also, it says 2102 when it is meant to be 2012.
Line 146-149 – I understand the reason for excluding time around home and work due to participants not turning off their devices but why was 125m decided upon? Was this based on some formative analyses? Average lot size?
Line 171 – were individuals observation time used to offset the duration estimate?
It would also be good to get an idea of how the travel logs worked. Did participants record the trip before they left or after the trip was concluded? Did they report every destination they visited or the primary reason?
Line 219 – A flow chart of participant exclusion with reasons may help bring the methods together – 712 potential participants 516 enrolled 18 excluded for missing demographic responses, 477 had concurrent GPS and travel log data. Sixteen removed for incomplete data etc.
Line 227 – perhaps a point for discussion that 30% of the visits could not be verified, given the existing limitations of self-report data, I think this needs to be discussed in more detail in the discussion with options for alternatives to self-reported measures of outcomes (visits).
Line 231 – do you know why the visits were not recorded by GPS? Was the data erroneous? Or the participant forgot to use the GPS?
Results
Could you combine tables 4-6 into a table like table 3 to make it a bit more readable.
Line 257- tables 46 is meant to be tables 4 to 6.
Line 262 In addition to comment about line 30, is it possible that the duration is more effective because it is more likely a reflection of participants walking (thus higher accessibility), while counts could reflect a large number of exposures by participants driving by, thus the results are bias by mode of transport?
Discussion:
Line 285 – Again I think duration may be associated with exposure because of points on line 30 and line 262. It would be good to discuss this in more detail.
Line 318-322 – Perhaps be more cautious and avoid using terms like to ‘affect their behaviour’ as you are only testing associations between your exposure measure and behaviour which are not casual and have methodological limitations.
Line 331 – again I feel comparing counts to duration may be an unfair comparison, given duration is likely reflect difference transport modes, leisure activity (shopping v commuting) and the outcome (visits to FFRs). I think this needs to be discussed further in order to justify that duration is a superior/alternative to counts.
Line 335 – This is were other measures could be used to validate unrecorded visits to FFRs such as ST DBSCAN, wearable cameras, amenities data or some combination of more objective measures.
References I referred to earlier are below:
Brusilovskiy, E., Klein, L.A., Salzer, M.S., 2016. Using global positioning systems to
study health-related mobility and participation. Soc. Sci. Med. 161, 134e142.
https://doi.org/10.1016/j.socscimed.2016.06.001.
Chambers, T., Pearson, A.L., Kawachi, I., Rzotkiewicz, Z., Stanley, J., Smith, M., Signal, L.,
2017. Kids in space: measuring children's residential neighborhoods and other
destinations using activity space GPS and wearable camera data. Soc. Sci. Med. 193,
41–50. https://doi.org/10.1016/j.socscimed.2017.09.046
Author Response
Reviewer comments are show in Courier 11 point, and our responses are shown in Calibri bold italic 11 point directly below each comment.
R | Thank you for the opportunity to review the manuscript entitled “A time-based objective measure of exposure to the food environment”. This paper used participant GPS data to use a new methodological measure of exposure to the food environment, duration. The paper demonstrates the need for a more nuanced approach to determining individual’s exposure to the food environment that extends beyond arbitrary boundaries using buffers around individual’s home and workplaces. Overall, I feel the paper makes a good contribution but there are aspects of the methods, given this is primarily a methodological paper, that need clarification and justification to highlight the utility of this approach. Further, I am not convinced on the relationship between the exposure (duration) and outcome (visits). These concerns are highlighted below in order of where they appear. |
Thank you for the very helpful review. | |
Abstract | |
R | Line 19- include the age of the participants. |
Change made | |
R | Line 30- My major concern is that people who visit a fast food restaurant would have substantially higher amounts of time spent in proximity to FFRs, thus the exposure and outcome are the same? |
We briefly mention this in Line 25, noting that we removed self-reported exposures from these measures. In lines 73-77 we use Chaix’s concept, selective mobility bias, to frame this issue. We revisit this issue again in the seventh paragraph of Section 4: Discussion (lines 349-356). We subtracted from our exposure measures time spent in the buffers when we were able to link that time to FFR trips reported in the Travel Diary. However, this is a problem for FFR visits that were unreported in the Travel Diary. See Chaix, B. et al., 2013. GPS tracking in neighborhood and health studies: A step forward for environmental exposure assessment, a step backward for causal inference? Health & place, 21C, pp.46–51. Available at: http://www.ncbi.nlm.nih.gov/pubmed/23425661 | |
Introduction | |
R | I think a larger discussion of advantages and limitations of travel logs is warranted in the introduction as your paper heavily relies on these. How many FFR visits do you feel were missed by relying on travel logs? Other studies have used individual-level GPS data to inform time-based measures of participants visits to destinations (Brusilovskiy et al., 2016) and children’s visits to food retails outlets (Chambers et al 2017 also used wearable cameras) using the ST DBSCAN algorithm (references provided at the bottom). |
Thanks for drawing our attention to the Brusilovskiy et al. and Chambers et al. papers. Both of these were mainly focused on automatically identifying visited destinations from passively collected data, whereas our main aim was to estimate time-weighted exposure to specific locations. Unfortunately, we have no way of knowing how many FFR visits were missed by relying on the travel logs. However, we were able to do the opposite and measure how many of the 273 visits to FFRs reported in the travel log had corresponding GPS data suggesting a visit. Applying a tolerance of +/- 10 minutes to visit times reported in the log resulted in 211 instances (79.5% of reported visits) in which a set of GPS points were inside the boundaries of a tax assessor parcel containing an FFR with a brand name that matching that of the FFR reported in the log. These findings were reported in Scully, J. Y., Vernez Moudon, A., Hurvitz, P. M., Aggarwal, A., Drewnowski, A., & Jacobs, D. (2017). GPS or travel diary: Comparing spatial and temporal characteristics of visits to fast food restaurants and supermarkets. PLOS ONE, 12(4), e0174859. https://doi.org/10.1371/journal.pone.0174859 It would be interesting as a separate study to “validate” FFR visits (and/or quantify visits that were missing from the travel diary using a method such as ST_DBSCAN. | |
Methods: | |
R | Lines 94 & 134 – What was the data completeness over those seven days? How many Erroneous GPS points were removed, how much of a problem was this? |
Our threshold of data completeness was 3 observation days. We removed 16 participants from the sample who did not meet this threshold due to improperly filled out travel logs or obvious errors in the GPS such as the total number of GPS points in the data totaling less than 100. We removed less than 2% of the GPS points from the sample using the PALMS, and Tsui and Shalaby data cleaning procedures. At the line segment level we only removed 0.14% of the line segments with durations longer than 30 seconds. | |
R | Line 95 – Was there any effect of seasonality in participants’ mobility patterns or FFR visits? Also, it says 2102 when it is meant to be 2012. |
2102 was changed to 2012—our apologies for this oversight. We did not check for seasonality in this study. Unfortunately, with a sample size of 412 we needed to limit our comparisons. | |
R | Line 146-149 – I understand the reason for excluding time around home and work due to participants not turning off their devices but why was 125m decided upon? Was this based on some formative analyses? Average lot size? |
We followed the precedent set by Hurvitz et al. 2014. Section 2.4 Exposure Measures modified to reflect this. Due to GPS drift, any given point has some error; although GPS manufacturers typically state 3-5 m, we found that >90% of GPS points were within 125 m of the parcel centroid (roughly a 1-block face radius in pre-WWII US neighborhoods) for a data set collected from a GPS unit at a stationary location within a typical Seattle area wood-frame house. Any exclusionary buffer will unfortunately include both false positives and false negatives; we thought that a larger buffer would be erring on the side of caution.
See Hurvitz, P. M., Moudon, A. V., Kang, B., Fesinmeyer, M. D., & Saelens, B. E. (2014). How far from home? The locations of physical activity in an urban U.S. setting. Preventive Medicine, 69, 181–186. https://doi.org/10.1016/j.ypmed.2014.08.034 | |
R | Line 171 – were individuals observation time used to offset the duration estimate? |
We did not apply a statistical offset in the logistic regression, but rather prepared the data so that the FFR exposure values were normalized to the person-day, and the dependent variable was binary (zero versus one or more FFR visits over the study period). Our understanding is that if we used raw counts of FFR visits or minutes of exposure, we would need to use the offset, which was not the case here. | |
R | It would also be good to get an idea of how the travel logs worked. Did participants record the trip before they left or after the trip was concluded? Did they report every destination they visited or the primary reason? |
A sentence responding to these questions was added to the first paragraph of Section 2: Material and Methods. | |
R | Line 219 – A flow chart of participant exclusion with reasons may help bring the methods together – 712 potential participants 516 enrolled 18 excluded for missing demographic responses, 477 had concurrent GPS and travel log data. Sixteen removed for incomplete data etc. |
We have included a figure with the exclusion criteria (Figure 1) | |
R | Line 227 – perhaps a point for discussion that 30% of the visits could not be verified, given the existing limitations of self-report data, I think this needs to be discussed in more detail in the discussion with options for alternatives to self-reported measures of outcomes (visits). Line 231 – do you know why the visits were not recorded by GPS? Was the data erroneous? Or the participant forgot to use the GPS? |
We discuss this issue in last two paragraphs of Section 4: Discussion. We are not sure how much more can be said on the topic. Unverified visits are black boxes. They could have been the result of self-report error or a failure to carry the GPS device. We have no way of assessing the reasons a self-reported visit would go unverified by GPS data. At this time, we think that research must use self-report and GPS in tandem. GPS could be used to imply that a visit had taken place simply because they measure XY locations, but self-report provides descriptive information about the location and the uses at that site. GPS points in proximity to an FFR may indicate a visit or a driver who has pulled over to take a phone call. Together, the two sources of locational data can be used to create a more complete picture. | |
Results | |
R | Could you combine tables 4-6 into a table like table 3 to make it a bit more readable. |
Thank you, the consolidated table makes much more sense. | |
R | Line 257- tables 46 is meant to be tables 4 to 6. |
Fixed by consolidating Tables 4 through 6 into one table. | |
R | Line 262 In addition to comment about line 30, is it possible that the duration is more effective because it is more likely a reflection of participants walking (thus higher accessibility), while counts could reflect a large number of exposures by participants driving by, thus the results are bias by mode of transport?
Line 285 – Again I think duration may be associated with exposure because of points on line 30 and line 262. It would be good to discuss this in more detail. |
A paragraph was added to Section 4: Discussion to address your concerns. | |
Discussion: | |
R | Line 318-322 – Perhaps be more cautious and avoid using terms like to ‘affect their behaviour’ as you are only testing associations between your exposure measure and behaviour which are not casual and have methodological limitations. |
Noted and changed. | |
R | Line 331 – again I feel comparing counts to duration may be an unfair comparison, given duration is likely reflect difference transport modes, leisure activity (shopping v commuting) and the outcome (visits to FFRs). I think this needs to be discussed further in order to justify that duration is a superior/alternative to counts. |
We hope that we managed to follow this advice in the new paragraph added to Section 4: Discussion. | |
R | Line 335 – This is were other measures could be used to validate unrecorded visits to FFRs such as ST DBSCAN, wearable cameras, amenities data or some combination of more objective measures. References I referred to earlier are below: |
These suggestions are valuable. Future studies would benefit by focusing on the validity of travel diaries and FFR utilization to include usage of wearable cameras, or using ST_DBSCAN as a way of estimating FFR visits that were not recorded in the diary. We have added three references supporting this (Chambers et al. 2017; Brusilovskiy et al. 2016; Carlson, J.A. et al., 2014). |
Round 2
Reviewer 1 Report
looks good to me - thanks to the authors for their hard work tidying up the minor comments
Reviewer 2 Report
The authors have addressed my concerns sufficiently.